# FDI and Wellbeing: A Key Node Analysis for Psychological Health in Response to COVID-19 Using Artificial Intelligence

**DOI:** 10.3390/ijerph20065164

**Published:** 2023-03-15

**Authors:** Da Huo, Jingtao Yi, Xiaotao Zhang, Shuang Meng, Yongchuan Chen, Rihui Ouyang, Ken Hung

**Affiliations:** 1School of International Trade and Economics, Central University of Finance and Economics, No. 39 South College Road, Haidian District, Beijing 100081, China; xiaotaozh@vip.sina.com (X.Z.); mengshuang@cufe.edu.cn (S.M.); 2School of Business, Remin University of China, No. 59 Zhongguancun Road, Haidian District, Beijing 100872, China; yijingtao@rmbs.edu.cn; 3School of Art, Sun Yat-Sen University, No. 135 West Xingang Road, Guangzhou 510275, China; chenych96@mail.sysu.edu.cn; 4China Center of Internet Economics, Central University of Finance and Economics, No. 39 South College Road, Haidian District, Beijing 100081, China; ouyangcass@163.com; 5A.R. Sanchez School of Business, Texas A&M International University, 5201 University Boulevard, Laredo, TX 78041, USA; ken.hung@tamiu.edu

**Keywords:** FDI from China, wellbeing, super-efficiency DEA, Tabu search, immune algorithm, key node analysis

## Abstract

Developing countries are primary destinations for FDI from emerging economies following the World Investment Report 2022, including destinations in OECD countries. Based on three theoretical lenses and case analyses, we argue that Chinese outward FDI has impacts on wellbeing in destination countries, and that this is an important issue for psychological health in response to COVID-19. Based on the super-efficiency DEA approach, our study investigated the impact of Chinese outward FDI on wellbeing in OECD countries. We also applied a Tabu search to identify country groups based on the relationship between Chinese outward FDI and wellbeing and we developed a key node analysis of the country groups using an immune algorithm. This research has implications for public administrators in global governance and could help shape FDI policies to improve psychological health of the destination countries in response to COVID-19.

## 1. Introduction

As a result of the economic reform and open-door policy in China, both Chinese companies and foreign investors in the domestic market have become increasingly competitive, allowing Chinese companies to compete in the global market [1]. According to the Ministry of Commerce of China, the Chinese outward FDI (foreign direct investment) in the global market has increased from USD 2.7 billion in 2002 to USD 196.15 billion in 2016 (Figure 1). Factors such as readily available cash, an artificially competitive cost of capital, and wage advantages have made China a competitive player in cross-border mergers and acquisitions [2]. The cross-border mergers and acquisitions can be observed in terms of Lenovo’s acquisition of IBMs PC unit, Shuanghui Group’s acquisition of Smithfield’s food processing business, and Midea’s acquisitions of Toshiba’s white goods. Emerging market companies, including Chinese companies, have become strategic asset-seeking investors and they seek to acquire both information and knowledge on how to operate internationally [3]. These international operations include Lenovo’s acquisition of the IBM PC division, as previously mentioned, and the acquisition of Volvo by the Geely Holding Group in 2010 [4].

In competition with and learning from companies from advanced countries, Chinese companies have increased their management and technical skills to the international standards required to enter developed markets [5]. The context of Chinese OFDI into the European Union (EU) attracted scholars to conduct research on OFDI from emerging economies to developed countries in international economics [6]. It was found that Chinese companies have expanded their investments from developing countries to developed regions since the 1990s, in favor of locations with offshoring sources and financial centers [7]. It was also found that there has been an increasing amount of FDI from China to the U.S. in recent years. Emerging economies such as China are playing an increasingly important role in the global market, and the current knowledge about the effect of OFDI from the emerging economy on the wellbeing of destination countries is inadequate [8]. As OECD countries are important destinations of Chinese OFDI, the extent to which Chinese FDI affects the wellbeing of OECD countries needs to be determined.

We investigated the effect of outward FDI from China on wellbeing in the OECD destination countries. Following case analyses as evidence of the effects of OFDI from China to destination countries, we further applied the super-efficiency data envelopment analysis, the Tabu search, and the key node analysis methods to analyze the different effects among OECD countries. This paper makes several contributions to the literature. Even though there are some existing studies on inward FDI on the host countries’ economic development, labor market, etc., we are among the first to study FDI flows’ effects on the wellbeing of the host countries. We analyze the mechanisms of how outward FDI from China affects wellbeing in OECD countries, representing developed destinations from social, organizational, and individual perspectives. Furthermore, we employ artificial intelligence to analyze the effects of FDI, which introduces an interdisciplinary approach to this line of research. This research applies the super-efficiency data envelopment analysis (DEA) approach to analyze the impact of FDI on wellbeing. This research further applied the artificial intelligence approach, including the Tabu search method, to identify the heuristic solution of OECD countries, targeting the shortest interconnections based on the impact of FDI flow on the wellbeing of destination countries. In addition, this research applies key node analyses based on the immune algorithm to destination countries according to the impact of FDI on wellbeing and reveals the heterogeneity of OECD country groups.

Psychological health is an important consideration in the recovery of manufacturing and service work in response to COVID-19, and the wellbeing of residents can be highly weighted in determinants of the public’s psychological health. This research offers an innovative vision to analyze the effect of FDI flow on wellbeing, aiming to lead an improvement in psychological health in recovery from COVID-19 based on a key node analysis of destination countries. This is achieved using artificial intelligence. This research imports an inter-disciplinary study investigating global governance in public health by acquiring economics and health analytics based on artificial intelligence. Our findings help to further reveal the channels applying the FDI from emerging markets to enhance the recovery of psychological health in response to COVID-19, with the support of upgraded analytics based on AI technology.

This research is an important contribution to the inter-disciplinary study of public health research, and it offers support to psychological health as part of the recovery of the manufacturing and service networks from COVID-19 based on artificial intelligence in computation science. This research investigates the effect of FDI as an economic issue related to wellbeing, influencing the psychological health of host countries. Public wellbeing offers important psychological health support based on the public’s life satisfaction, including factors such as health, safety, happiness, job, housing, etc. Psychological health is an important issue to consider during recovery from COVID 19. The psychological experience of life satisfaction can assist the public health administration in the recovery of manufacturing and service work in the wake of COVID-19. This study analyzes the effect of economic issues on psychological health using artificial intelligence and aims to initiate an inter-disciplinary framework to support the recovery from COVID-19 from a public psychological health perspective.

## 2. Literature Review and Case Evidence

### 2.1. Literature Review

This research initiates a multi-level theoretical framework to analyze the effect of FDI on wellbeing in destination countries at social, organizational, and individual levels, as shown in Figure 2. Psychological health is an important concern for public health administration in relation to COVID-19 recovery [9]. Analyzing the effect of economic sectors on wellbeing can offer important support to enhance public psychological health. The integrative function of FDI on wellbeing generates an organic system in public administration of psychological health in response to COVID-19 at individual, organizational, and social levels, furthering the inter-disciplinary system of public health administration’s acquisition of an economic and psychological vision.

#### 2.1.1. Social Level Analysis: Social Influence Theory

The motivators for wellbeing at the social level can be demonstrated by social responsibility and localization of investment in host markets [10]. Social connections and the sense of corporate responsibility as a global citizen can motivate both employees and residents [11]. Social responsibility can motivate wellbeing by contributing to regional development, such as the “One Thousand Dream” program by Huawei in the CEE region. The stability of corporate management motivates the wellbeing of employees and residents, who provide organizational commitment and continuous support to corporate businesses such as Midea investment in Japan by supporting current business operations. The localization of market operations motivates wellbeing by encouraging social communication and interaction with employees and residents, such as localized cooperation between Leveno and Italy NGO.

According to the social influence theory, social influence is important, as it shapes individual behavior [12]. Formal and informal social norms of accepted behavior can influence an individual’s belonging to a group. MNEs are regarded as key players in social responsibility in the destination countries, given their global influence and activities [13]. Following the social influence theory, social responsibility posits that investors who are involved in destination markets can improve the positive response of local people to foreign investors. The localization of investment in foreign markets also facilitates the engagement of foreign investors in connection with local people.

Many kinds of literature have proved that FDI flows improve the life-relevant indexes in the destination countries. It was found that investment price level is an important cross-country determinant of wellbeing [14]. Meanwhile, FDI inflows motivate the GDP growth rate, growth in industry and service sectors, reduction in unemployment, reduction in poverty, improvement in the standard of living, an increase in foreign exchange reserves, an increase in exports, and an improvement in the stock market [15]. The quality of life, urban aesthetic, and local development policies are important qualitative soft factors for business establishment and FDI choices in potential locations, and therefore countries can be motivated to improve quality of life in order to attract FDI flows [16]. It was found that FDI inflows have significantly contributed to poverty reduction in African countries, and that better institutional quality and human capital development, as well as better functioning financial systems with FDI, are associated with reducing poverty [17]. Favorable social policies referring to medical, housing, and social services are important to the wellbeing of immigrant residents. Cultural and communication barriers can affect the wellbeing of immigrant residents [18]. FDI flows can also encourage policy reform, industrial development, reduction in political risks, the unemployment rate, economic growth, and macroeconomic performance. FDI can be facilitated by improving the investment environment through modified macroeconomic policies, strengthened institutions intensification of structural reforms, rapid liberalization, and market regulation [19]. It was found that encouragement of FDI inflows will further promote ICT industrial development in the country [20].

Therefore, based on the macro-social level analysis, we argue that outward FDI from China may have an impact on wellbeing in the host countries.

#### 2.1.2. Organizational-Level Analysis: Incentive Compatibility Theory

The motivators of wellbeing at the organizational level are presented by sustainability and survival in the following cross-case study. Organizational development can be an incentive to encourage future expectations of employees’ wellbeing [21]. Sustainability motivates wellbeing through enhanced R&D activities, such as Geely’s investment in Sweden. Competitiveness can also motivate wellbeing using enhanced technology, as demonstrated by Huawei’s investment in Latin America. Survival can motivate wellbeing by providing support to current business and their ventures, such as Midea’s investment in Japan. The expansion motivates wellbeing by providing a broadened coverage of business deals, such as in Shuanghui’s investment in the U.S.

In the incentive compatibility theory, individuals follow the rules of an organization when pursuing certain achievements [22]. The compatibility of incentives allowing individual members to achieve their targets by following the rules may be important to the organizational process. Following incentive compatibility theory, employees’ motivation can be enhanced by further recognition and acceptance of organizational missions and objectives [23]. The enhanced sustainability of corporate development by FDI can improve the positive feelings of local people. Additionally, the strengthened survival of corporate business as a result of support from foreign investors contributes to the positive prospects of local people [24].

FDI also has an important impact on corporate behavior in cross-border business activities and influences the wellbeing of employees. Corporate strategies and business decisions can be influenced by the FDI in cross-border activities [25]. It was found that the location of FDI is of great importance to the firm’s performance of multinational companies [26]. The enhancement of OFDI also offers support to innovation at home market [27]. It was found that high-commitment entry is positively associated with the affiliate being located in areas with strong economic, cultural, and historic links with the parent company [28]. It was also found that China influences the African countries through the diffusion of its management ideas and skills via soft power, which may subsequently improve wellbeing [29].

Therefore, based on the organization-level analysis, we argue that outward FDI from China may improve the organization in the host countries, leading to an increase in wellbeing.

**Figure 2 ijerph-20-05164-f002:**
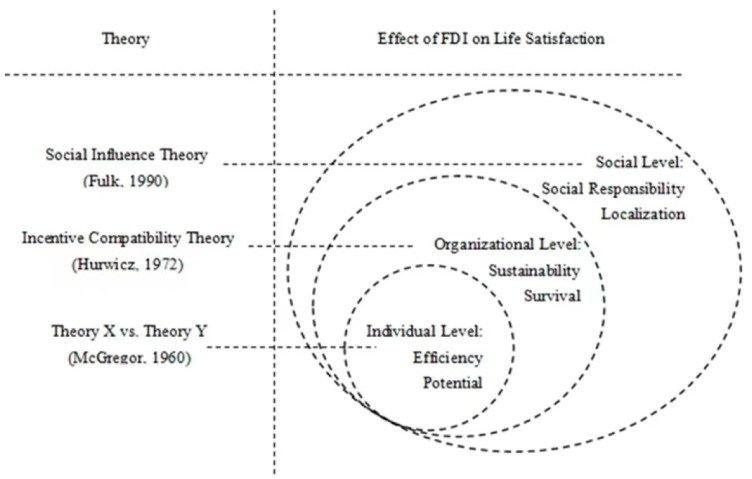
Theoretical explanation of the effect of FDI on wellbeing at different levels [12,22,30].

#### 2.1.3. Individual-Level Analysis: Theory X and Theory Y

According to Theory X, employees need supervision due to limited self-directed and autonomous behavior when solving organizational problems. The motivators that help individuals to complete their tasks to a higher standard and achieve greater efficiency are a vital part of enhancing their performance to achieve higher physiological rewards, which contribute to wellbeing. Greater efficiency can motivate wellbeing via enhanced work achievement and time-saving experiences, as demonstrated by Lenovo’s investment in Italy, which led to higher efficiency in engineering design. Innovativeness can motivate wellbeing by enhancing employees’ competencies in their roles, as demonstrated by Huawei’s investment in CEE, which improved the digital experience of its younger employees.

According to Theory Y, management work can be developed by helping employees to further explore their potential instead of being dependent on the control and commands of others [30]. The motivators that contribute to the future development of employees can improve wellbeing, and these business opportunities also enhance the wellbeing of employees with promising psychological expectations. Additional training and experience offers employees a higher level of confidence, providing them with tools for future development and subsequently enhancing their wellbeing [31]. Examples of this include Changhong’s investment in the Czech Republic, offering informatization in quality management, and Chinese companies’ investment in Germany, which provided employees with additional training. These contributions can motivate employees to participate in future contributions to corporate operations, such as Changhong’s investment in the Czech Republic, encouraging local employees to work with the corporate manager and develop solutions to improve their efficiency at work.

Theory X explains the motivation of workers based on the satisfaction of the physiological needs of those with fixed job responsibilities, and Theory Y interprets the motivation of workers based on fulfillment of the psychological needs of those with flexible job positions that encourage workers to contribute and further develop their professional potential [32]. According to Theory X, improving working efficiency and job responsibility offers employees a higher level of performance in return for career-related rewards and lower costs. The efficiency enhanced by foreign investment in business solutions and technical support can be an important source of motivation for employees. According to Theory Y, the motivation of employees as a result of foreign investment in further exploration of individual potential can also enhance the psychological satisfaction of employees with values of self-actualization. The potential development by foreign investment in professional training and participative teamwork involvement can offer a source of motivation to employees [33].

It was found that the development of individual wellbeing is also relevant to organizational behaviors in a business context [34]. While the MNEs initially used the natural assets of the country, their activities resulted in a higher-quality workforce, demonstrating more value-added, skill-intensive activities, better institutional quality, and human capital development-all of which are associated with reducing poverty [35]. Wellbeing at the individual level can be further influenced by FDI based on corporate strategies and business decisions. It was also found that the wellbeing of individuals in organizations can be influenced by the systematic work of the employee, as well as the overall organizational performance [36]. FDI can facilitate the flow of knowledge through the spill-over effect in business activities. Transferring knowledge within the organization can also help to develop the wellbeing of employees based on boundary-spanning activities [37]. It was found that wellbeing is influenced not only by a person’s wellbeing in comparison to that of their home country but also by their wellbeing compared to that of other countries around the world [38]. The motivation and justice of managerial behavior in an organization can be important to the wellbeing of employees [39]. The managerial behavior of companies in FDI in the global market can play an important role in the further enhancement of individual wellbeing in organizations based on global standards of managerial mechanisms. Meanwhile, it was found that the family–work conflict and autonomy satisfaction also affect managers’ wellbeing [40].

Therefore, based on the micro individual-level analysis, we argue that outward FDI from China can influence wellbeing in the host countries. The key node analysis of urban life can offer important support to public health administration in response to COVID-19 [41]. The inter-disciplinary vision from economics and health studies on the effect of FDI on wellbeing reveals an important channel to further enhance psychological health, aiding the recovery of manufacturing and service workers in the wake of COVID-19. Meanwhile, the application of artificial intelligence in the analysis of key nodes in country groups helps us to further understand the effective enhancement of psychological health by supporting the FDI outward flow from emerging markets.

### 2.2. Evidence by Case Study

Before conducting the theoretical analyses, we applied case studies to develop a detailed understanding of the effect of FDI on wellbeing in the destination countries.

(1)Huawei’s investment in Latin America: Huawei plans to increase its involvement in the development of data storage and 5G network services in the Latin American market. The cloud technology and blockchain services operated by Huawei offer further technical support of digital management work in Latin America (resources from https://www.sohu.com/a/320468712_114774 (accessed on 14 June 2019)).(2)Huawei’s investment in Central and Eastern Europe (CEE): Huawei initiated an education program in the CEE area, which is called the “One Thousand Dream” program. This program offers free training to one thousand professional employees, donates one thousand books to libraries in each country, and organizes the donation of one thousand toys to children’s hospitals at local markets. This program aims to improve the digital experience of young people in CEE and motivates young people to carry out further exploration of the scientific area (resources from https://www.huawei.com/cn/news/2019/4/huawei-ict-one-thousand-dreams-program-europe (accessed on 12 April 2019)).(3)Geely’s investment in Sweden: Geely intends to initiate investment in Sweden to achieve further development of the Geely innovation center in Europe. The Geely innovation center, which is to be located in Lindgolm Technological Park, will further research on auto power, as well as sales and marketing groups for new vehicle models. Geely has offered contributions to economics and employment in Sweden and motivates exportation with synergies in R&D across China and Sweden markets (resources from http://auto.people.com.cn/n1/2017/0629/c1005-29370401.html (accessed on 29 June 2017)).(4)Chinese investment in Germany: Local government incentives encourage investments by providing enhanced employee revenue and R&D activities from companies involved in cooperation with FDI flow, in some cases even offering free training to employees in companies with foreign investment (resources from http://www.cgcpa.org.cn/jituanxinxikanwu/kuaguojingying/2013-08-26/3510.html (accessed on 26 August 2013)).(5)Midea’s investment in Japan: Midea acquired the white goods department of Toshiba company and now offers support to the continuous development of the Toshiba company from Japan. Midea has provided consistent support of branding, technology, marketing, and personnel functions of Toshiba since the acquisition, and its independent management system offers a powerful commitment to Toshiba with a stationary development of the global market based on current business operations. This mode of acquisition helps investor companies to overcome the hemolytic reaction in cooperation with acquired companies (resources from https://www.sohu.com/a/101207144_119737 (accessed on 4 July 2016)).(6)Lenovo’s investment in Italy: Lenovo has enhanced the investment in Italy with resource allocation after the acquisition of Thinkpad from IBM. In channel management, Lenovo has developed partnerships in cooperation with 1500 companies, and has also developed online distribution channels. The High-Performance Computing (HPC) department of Lenovo has long been involved in cooperation with the Italian NGO CINECA in support of scientific research in Europe, including the development of artificial intelligence. The HPC department of Lenovo offers powerful support to local services, such as enhanced efficiency in engineering design work for the Italian automobile manufacturer, Dallara (resources from http://finance.sina.com.cn/chanjing/gsnews/2020-01-17/doc-iihnzhha2951943.shtml (accessed on 17 January 2020)).(7)Shuanghui’s investment in the USA: the Shuanghui Group acquired Smithfield in the USA and offered to maintain the operation and management group of Smithfield. Shuanghui Group company also committed to preventing manufacturing shutdowns and employment cuts. The coverage of complete industrial chain by the Shuanghui Group following this acquisition offers both companies a good opportunity to stay in a high-end segment of the food processing market (resources from http://vip.stock.finance.sina.com.cn/q/go.php/vReport_Show/kind/search/rptid/1879279/index.phtml (accessed on 30 May 2013).(8)Changhong’s investment in the Czech Republic: Changhong Electronics has been further involved in market expansion in the Czech Republic. The Changhong company works with engineers from the Czech Republic to find solutions that allow improved management efficiency in assembly lines. Changhong launched self-innovation to facilitate the informatization of manufacturing and sales work. The combination of informatization in quality management and the localization of market operations highly speeds up the efficiency of the management work carried out by the Changhong company and offers support to solutions in labor cost control to the labor-intensive electronics manufacturing industry in the higher wage level European market (resources from https://epaper.gmw.cn/gmrb/html/2015-12/19/nw.D110000gmrb_20151219_4-08.htm (accessed on 19 December 2015).

Based on the case analyses, we categorized the effects that enhance wellbeing in destination countries into the social, organizational, and individual levels. Table 1 shows coding in the cross-case study regarding the effect of FDI on wellbeing. The case analyses provide some evidence in support of the effects of Chinese outward FDI in destination countries.

## 3. Data and Methods

Based on the case evidence and theoretical analyses, we used the DEA approach, Tabu Search, and key node analysis to empirically explore the effects of FDI on wellbeing in OECD countries and the heterogeneity among countries. The advancement of newly developed technology has enhanced the traditional managerial work, such as the application of block chain in upgrading of traceability of original achievement [42]. The advancement of technology further offers support to public health administration, such as the application of block chain to enhance traceability of infectious diseases [43]. This is an initiative study that leads a new vision for public administration and psychological health in response to COVID-19 based on inter-disciplinary study of economics and public health with the support of artificial intelligence technology. The discussion about multi-level effect of FDI on wellbeing reveals public administration for psychological health in recovery from COVID-19 at the individual, organizational and social level by involvement of FDI from emerging market. The advancement of super-efficiency DEA analysis enhanced the study of the effect of FDI on wellbeing, and the out-performance of Tabu search aiming at global heuristic solution offers further support to identify the country groups in global network externality. The key node analysis based on immune algorithm offers a creative analytical framework for practical solutions in public health and psychological health in response to COVID-19. This research will further our understanding of the development of the social health ecosystem based on psychological behavior in response to COVID-19, with the support of artificial intelligence.

### 3.1. Data

We have three main data sources. The Chinese outward FDI data were obtained from the 2016 Statistical Bulletin of China’s Outward Foreign Direct Investment. The GDP and population of OECD countries in 2016 were obtained from the World Bank database. The cultural indexes were from the Geert–Hofstede database. The historical effect of investment was involved through analysis of both FDI stock and FDI flow data sources. The period of the database was set to 2016, prior to the global trade disputes following unilateralism.

Wellbeing data and life relevant indexes were obtained from the OECD better life index database. The life-relevant indexes include general life satisfaction, housing, income, jobs, community, education, environment, civic engagement, health, happiness, safety, and work–life balance. OECD countries include Australia, Austria, Belgium, Brazil, Canada, Chile, Czech, Republic, Denmark, Finland, France, Germany, Greece, Hungary, Ireland, Italy, Japan, Mexico, the Netherlands, New Zealand, Russia, Sweden, Switzerland, the UK, and the USA. Table 2 shows the descriptive statistics of wellbeing in OECD countries of Chinese outward FDI. The Chinese outward FDI to OECD countries in 2016 is shown in Figure 3. The USA, Australia, Canada, Germany, France, the UK, and Russia are OECD destination countries with higher value of Chinese outward FDI in 2016. Additionally, Chinese outward FDI stock for OECD countries in 2016 is shown in Figure 4. The USA, Australia, the Netherlands, the UK, Russia, Canada, Germany, and France are OECD destination countries with higher values of Chinese outward FDI stock.

### 3.2. Methods

This research develops the study of public health administration in response to COVID-19 in the global network using artificial intelligence technology. The super-efficiency DEA model offers support to study the effect of FDI from the emerging market on the public’s wellbeing in destination countries, while the Tabu search offers efficient solutions for identifying the country groups in psychological health administration based on global heuristic solutions. The key node analysis based on immune algorithm can help reveal the group characteristics based on the study of key node countries in public health administration.

#### 3.2.1. Super-Efficiency Data Envelopment Analysis

The DEA approach has seen extensive use in operational research or management science [44,45], but appears less prevalent in international management fields [46]. DEA is used to identify investment efficiencies [47,48]. Following the work of Andersen & Petersen (1993), we apply the super-efficiency DEA (SDEA) approach and the slack model for analysis [49]. The functions are as follows:(1)$$\begin{array}{l} \min_{\theta ,\lambda }\;\theta \\ s.t.-q_{i}+Q\lambda \geq 0\\ \theta x_{i}-X\lambda \geq 0\\ I_{Ix1}^{\prime }\lambda =1\\ \lambda \geq 0 \end{array}$$

The $q_{i}$ is the output of wellbeing variables in OECD destination countries. $x_{i}$ is the input of Chinese FDI outflows to OECD countries (as the short-term input) and Chinese OFDI stock (as the long-term input) in OECD countries in 2016. Both Chinese FDI outflow and Chinese FDI stock to OECD countries are transformed by the logarithm. *θ* is the technological efficiency of Chinese OFDI based on the wellbeing in OECD destination countries. In comparison to constant return of scale, the variable return of scale adds a convex restraint that $I_{Ix1}^{\prime }\lambda =1$, so the decision units are compared to other units with similar scale when structuring the frontier.

Andersen and Petersen (1993) suggested that the super-efficiency DEA (SDEA) model can be used to measure the decision units that have an efficiency greater than one. The decision units are not compared to themselves when the frontier is constructed, and thus the frontier is structured by other decision units while the efficiency is measured. Figure 5 shows the adjustment in measurement of efficiency of SDEA based on output orientation with two sectors of inputs. The frontier in efficiency measurement is structured by efficient point B and C, without point A, as the case that it is compared to itself is excluded when structuring the frontier. Thus, efficient point A* lies between B and C, and point A is positioned over the efficient point A*. Therefore, the efficiency of point A can be greater than one, and the super efficiency of the DEA model is measured.

Based on the SDEA approach, two variables including constant return of scale efficiency (CRSE) and variable return of scale efficiency (VRSE) are calculated. The CRSE measures the efficiency in cases in which each entity is involved in a performance with maximum efficiency [50]. The VRSE measures the efficiency in cases in which entities are not guaranteed to perform at maximum efficiency, and entities efficiencies are compared to others with similar scales [51]. These two variables reflect two dimensions of the impact of Chinese outward FDI on wellbeing, and then they are used in the next steps to determine the key nodes.

#### 3.2.2. Tabu Search

Next, to investigate the interconnections of countries, we first calculate the Euclidean distances of OECD destination countries based on dimensions including Chinese OFDI flows and Chinese OFDI stock in the specific country, CRSE, VRSE, GDP, population, and cultural distances between China and OECD destination countries. The cultural distance is measured following Kogut and Singh (1988) based on Hofstede cultural indexes [52], including individualism, power distance, masculinity, uncertainty avoidance, pragmatism, and indulgence. To further identify the interconnections of OECD countries in terms of the efficiency of China’s OFDI on wellbeing, we apply the Tabu search approach.

The different groups of OECD countries are identified by a Tabu search that targets the shortest Euclidean distance across different OECD countries of Chinese OFDI. The function of the Euclidean distance is as follows:(2)$$d_{ij}=\sqrt{\sum_{k=1}^{p}{\left(x_{ik}-x_{jk}\right)}^{2}}$$
where $d_{ij}$ is the Euclidean distance between country i and country j. $x_{ik}$ is dimension k of country i, and $x_{jk}$ is dimension k of country j.

Glover (1986) initiated the use of the Tabu search method in artificial intelligence and operational research [53]. Tabu search is an efficient method that obtains the global solution to complex heuristic problems. Tabu search is an iterative method which has proven to be highly effective for solving the problems of clustering based on distance measures [54]. Using the Tabu search method, we can divide the sample into groups. Within each group, there is the shortest Euclidean distance among the members, which indicates that the group members share similar characteristics.

The algorithm used to identify the heuristic solution obtained via the Tabu Search is shown in Table 3. The solutions concentrate on the positioning of a specific unit in the program, which can be identified as a group of neighborhood solutions. For example, the solution to position of decision unit Country 1 can be switched with the position of the decision unit Country 2, and the sequence in the solution can be changed from 1, 2, 3, … 24 to 2, 1, 3, … 24. Furthermore, the position of Country 1 can be switched with the position of Country 3, and the solution can be changed to 3, 2, 1, … 24. Finally, the position of Country 1 is switched with the position of Country 24, and the solution is further changed to 24, 2, 3, … 1. The solutions that concentrate on the switch of position of Country 1 can be grouped as a neighborhood solution. Additionally, the algorithm structures a Tabu list that identifies the optimal solution for a neighborhood group. The candidate solutions are compared to optimal solutions, further identifying the heuristic solution so that the algorithm is not involved in local solutions.

Therefore, the Tabu search is performed to investigate the interconnections of OECD countries with the shortest interconnections in a group.

#### 3.2.3. Key Node Analysis

Finally, the immune algorithm is used to identify the key nodes that determine the effect of Chinese OFDI on the wellbeing of OECD countries. The immune algorithm can be applied to analyze the key nodes of the coding system based on the ratio of variables with which the country units are involved in the same quantile categories in comparison to the total. The key nodes of different groups can be identified so that the country nodes at key node positions can be further analyzed in relation to FDI and wellbeing.

The immune algorithm is an intelligence technology that targets heuristic solutions following the biological system based on the fitness of the antibody and antigen, as well as the concentration of antibodies. The function of the immune algorithm is as follows:(3)$$\begin{array}{l} A=\frac{1}{F}=\frac{1}{\sum d_{ij}Z_{ij}}\\ C=\frac{1}{N}\sum S\\ Exp=\beta \frac{A_{i}}{\sum A_{i}}+(1-\beta )\frac{C_{i}}{\sum C_{i}} \end{array}$$
where A represents the fitness of an antibody and antigen, and it is identified by the distance $d_{ij}$ and connection structure of different country nodes $Z_{ij}$. C represents the concentration of antibodies, and it is identified by similarities of antibodies S based on the ratio of similar coding in the antibody. Exp represents the expected reproduction; it is decided based on the fitness between antibody and antigen A, and the concentration of antibody C. The expected reproduction is encouraged in the biological system.

The variables that are involved in calculating the effect of FDI on wellbeing are coded by quantiles. The result of Tabu Search is further acquired using the coding system of the immune algorithm. The immune algorithm in key node analysis of country nodes is shown in Table 4. The first row represents the coding system of the UK, and the second row represents the coding system of the USA. The group characteristics are presented by Group, as the UK is involved in Group 2, and the USA is involved in Group 3. Other geographic variables are also coded, and the quantiles of FDI inflow, FDI stock, and GDP level of the UK and USA are in the same quantiles, with other variables at different quantiles.

## 4. Results

This research develops a leading study to investigate the effects of FDI on wellbeing using a super-efficiency DEA model and identifies the country groups via a Tabu search. The analytical result is further involved in key node analysis for psychological health in response to COVID-19 using an immune algorithm. The systematic analysis of wellbeing by the vision of FDI from the emerging market offers a new perspective to further the public administration of psychological health in recovery from COVID-19 based on artificial intelligence.

Figure 6 shows the distribution of CRSE and VRSE according to the SDEA approach, which reflects the impact of Chinese outward FDI on wellbeing in the destination countries. Australia, Brazil, France, Germany, Italy, Japan, the Netherlands, Russia, the UK, and the USA are at a higher level of investment efficiencies in terms of CRSE and VRSE, which indicates that the Chinese outward FDIs have greater impacts on wellbeing in these countries.

Table 5 shows the grouping of OECD countries based on a Tabu search that targets the shortest Euclidean distances across OECD destination countries. Finland, Ireland, Chile, Mexico, Hungary, Austria, Sweden, and New Zealand are in Group 1. Canada, Germany, France, Japan, Brazil, Italy, Russia, and the UK are in Group 2. The Netherlands, Australia, the USA, Greece, Czech Republic, Switzerland, Belgium, and Denmark are in the Group 3. Countries in the same group share similar characteristics of investment efficiencies and countries attributes.

The results of the simulation of the effect of Chinese outward FDI on wellbeing in Group 1, 2, and 3 based on a neural network is visualized in Figure 7, Figure 8 and Figure 9, respectively. It shows that Group 1 is distributed at a higher concentration of lower VRSE, and Group 2 is distributed at a higher concentration of higher CRSE. However, Group 3 is divergent in CRSE and VRSE.

Within the three groups, Group 1 has a medium value of VRSE and CRSE. We find that the countries in Group 1 have a limited population; therefore, they still have the potential to explore the Chinese FDI investment efficiency. Group 2 has a high value for both VRSE and CRSE. We find that countries in Group 2 have advanced economic development and competitive technology; thus, they have higher investment efficiency. However, Group 3 has two types: one with low value for both VRSE and CRSE and another with low value for VRSE but a high value for CRSE. Since the countries in the same group have the shortest distance between them, these countries have more spillover with one another. Thus, using the information from the groups, one can learn how to increase the efficiency within each group based on the spillover effects.

The key nodes analysis carried out using the immune algorithm to determine the effect of Chinese outward FDI on wellbeing is shown in Figure 10. The key nodes in the effect of Chinese outward FDI on the wellbeing of OECD countries are from New Zealand, the UK, and Greece. To identify the key nodes in the network, one can learn from the key node to derive the characteristics of the group.

## 5. Conclusions

With the development of the Chinese economy and the open-door policy, Chinese OFDI is increasing in many destinations, including OECD countries. This study finds that Chinese OFDI affects the wellbeing in OECD destination countries. The determinants of wellbeing arising from FDI can be categorized into the social, organizational, and individual level. Following the social influence theory, the social responsibility, localization of business operation, and stability of corporate management of MNEs contribute at the social level. Following incentive compatibility theory, MNEs sustainability, competitiveness, survival, and expansion contribute at the organizational level. Following Theory X and Theory Y, MNEs innovativeness, efficiency, and potential offer support at the individual level. This research explores an incentive compatible channel that takes advantage of the positive externality of cross-border investment in the development of public health administration in the host country, by acquiring the development of life satisfaction based on health, job, income, etc., to support people’s psychological wellbeing during the recovery from COVID-19.

Based on the above analysis, this study identifies the impacts of Chinese outward FDI on wellbeing in OECD countries via super-efficiency data envelopment analysis. We find that Australia, Brazil, France, Germany, Italy, Japan, the Netherlands, Russia, the UK, and the USA exhibit a higher investment efficiency, which indicates that Chinese FDIs have larger impacts in these countries.

We further identify the interconnections of OECD countries using the Tabu search method. Based on the impact of Chinese outward FDI on wellbeing, cultural distances and country attributes, we can separate the countries into three groups and identify the shortest Euclidean distances with the group. We find that Finland, Ireland, Chile, Mexico, Hungary, Austria, Sweden, and New Zealand are in Group 1. Canada, Germany, France, Japan, Brazil, Italy, Russia, and the UK are in Group 2. The Netherlands, Australia, the USA, Greece, Czech Republic, Switzerland, Belgium, and Denmark are in Group 3. Using a simulation analysis, we find that countries from Group 1 hold consistent CRSE and VRSE, which indicates that the investment efficiencies have the potential to improve. The countries from Group 2 are both high in CRSE and VRSE, which indicates that the investment efficiency is quite high in this group. The countries from Group 3 are in two sub divergence: one for countries with a lower VRSE and higher CRSE, while the other with low CRSE and VRSE.

Finally, the key node analysis is performed to study the effect of Chinese outward FDI on the wellbeing of OECD countries. We find that the key nodes used to describe the effect of Chinese outward FDI on the wellbeing of OECD countries are from New Zealand, the UK, and Greece. The key nodes represent the characteristics of CRS and VRS in three groups. New Zealand has a relatively higher level of CRS and a median level of VRS. The wellbeing of New Zealand is at a high level in the global market, but the understanding and psychological feeling of wellbeing of New Zealand can be further improved. The investment to countries in Group 1 can further develop the psychological understanding of the wellbeing at an individual level. The UK is high at both the CRS and VRS level. The satisfaction of health conditions, employment, family, and psychological needs contribute to wellbeing in the UK. The investment in countries in Group 2 can offer support for work–life quality and social life improvement in the development of wellbeing at the social level. Greece is relatively lower in terms of its CRS and VRS level. Due to the negative influence of economic recession, the shrink of social benefit packages and the hit to the physical economy affect wellbeing. The investment in countries in Group 3 can be further involved in economic development and the improvement of life benefits for wellbeing at the organizational level.

Following the findings of this research, the host country can take advantage of foreign direct investment from the emerging market to support public administration in relation to psychological health during recovery from COVID-19. This can be carried out at the social, organizational, and individual level. The social services and security support can help to enhance psychological wellbeing at the social level. The enhanced business communication and economic development can offer support to improve psychological wellbeing at the organizational level. The understanding and feelings of individuals can be considered in order to provide comfort and improve psychological wellbeing at the individual level. The integrative public administration for psychological health at a multi-level function based on positive external effects of foreign direct investment offers important support to the global governance of public psychological health during recovery from COVID-19.

This study has significant implications for public administrators in global governance, as it reveals the effect of FDI on wellbeing in the host countries. We should notice the spillover effects within each group and the key characteristics in the key nodes. The practical solutions following the key node analysis based on artificial intelligence can help us to develop positive network externality in public health administration for country groups in response to COVID-19. Based on the key node analysis using artificial intelligence, it is suggested that countries in Group 1 can further enhance the individual feeling of wellbeing for manufacturing and service workers recovering from COVID-19 based on the FDI flow from emerging markets; for example, by maintaining a stable medicine supply and Medicare support, aiming to enhance the concern surrounding psychological health in individual life. It is also suggested that countries in Group 2 can take advantage of social security systems in support of post-COVID-19 services at the social level by working with the FDI flow from emerging markets. This includes the facilitation of medical services with advanced Medicare infrastructure, aiming to improve experiences of psychological health in relation to people’s social lives. It is further suggested that countries in Group 3 can support organizations’ survival and sustainable development during recovery from COVID-19 by working with the FDI flow from emerging markets; for example, by offering employment and investment opportunities, aiming to enhance psychological health support at the organizational level. Therefore, revealing the effects of the group characteristics of FDI on wellbeing using artificial intelligence may be helpful to public administrators, furthering the development of economic and health systems that are used to support psychological health in response to COVID-19.

This is an inspiring research work that initiates the study of public health administration in response to COVID-19 from a psychological perspective by acquiring social science and computation science areas. The development of an organic system for public health administration in acquiring economics and psychological study contributes to the inter-disciplinary literature of multi-level global governance at the individual, organizational, and social level. The application of artificial intelligence to practical solutions of public health administration from the psychological perspective in response to COVID-19 leads to an advancement of analytical frameworks in global governance based on social science and computation science studies. Additionally, this was a study of public administration for psychological health based on the economic sector’s effects on wellbeing. These effects were determined using artificial intelligence. This research was based on evidence of FDI from emerging economies to developed countries, with FDI data sources from China, and it used a wellbeing database from OECD countries. Further studies on global health administration following this analytical framework can be developed.

## Figures and Tables

**Figure 1 ijerph-20-05164-f001:**
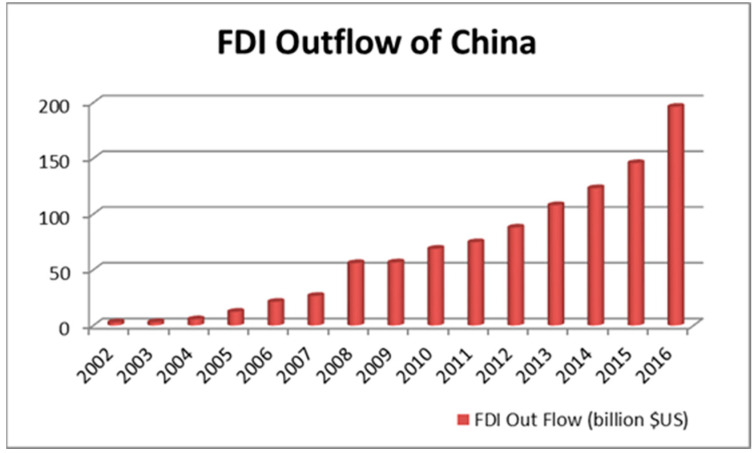
Chinese OFDI from 2002 to 2016.

**Figure 3 ijerph-20-05164-f003:**
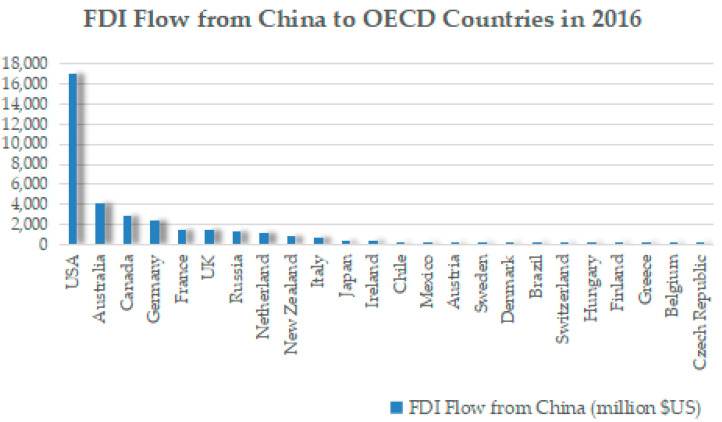
Chinese outward FDI flow to OECD countries in 2016.

**Figure 4 ijerph-20-05164-f004:**
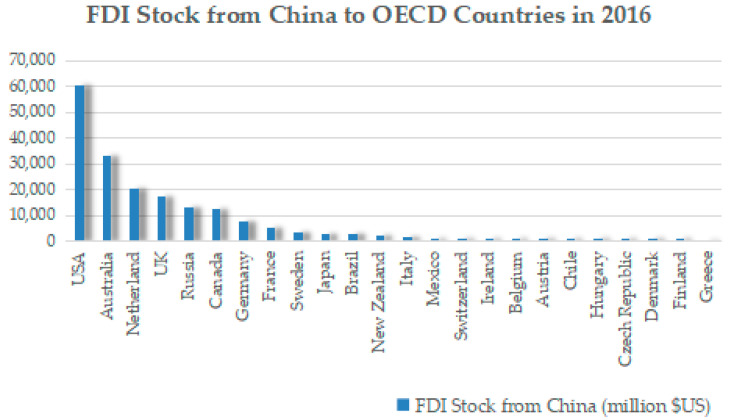
Chinese outward FDI stock to OECD countries in 2016.

**Figure 5 ijerph-20-05164-f005:**
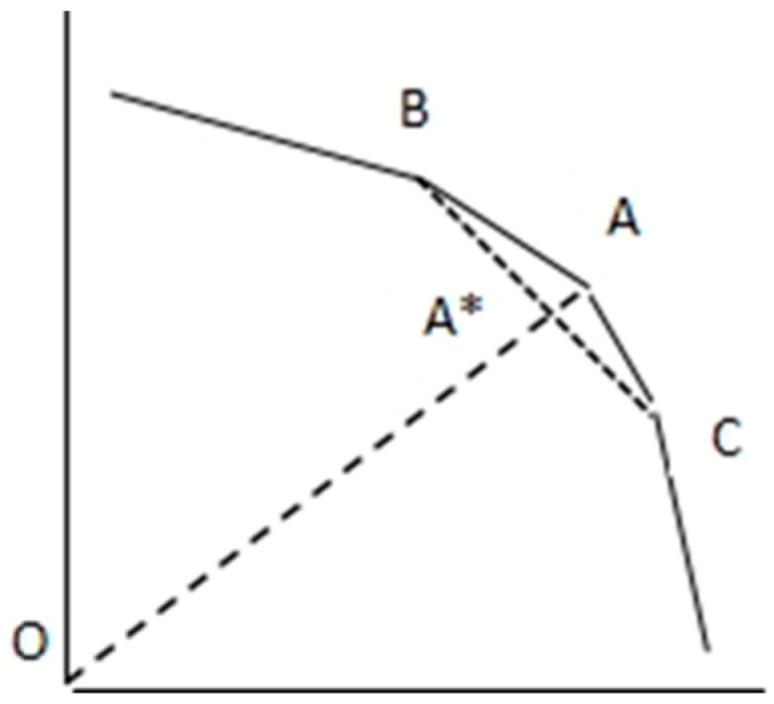
The super-efficiency data envelopment analysis. A*: efficient point.

**Figure 6 ijerph-20-05164-f006:**
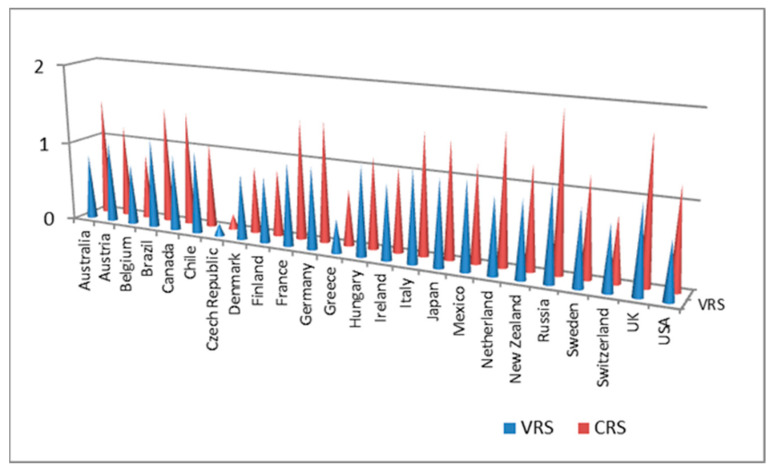
Investment efficiency of Chinese outward FDI on wellbeing from SDEA approach.

**Figure 7 ijerph-20-05164-f007:**
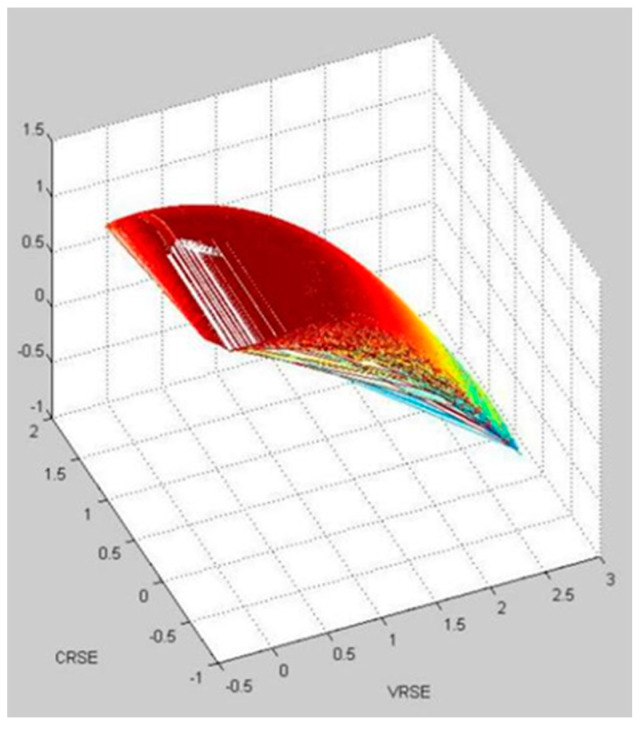
The visualization of simulation of the effect of Chinese outward FDI on wellbeing in Group 1.

**Figure 8 ijerph-20-05164-f008:**
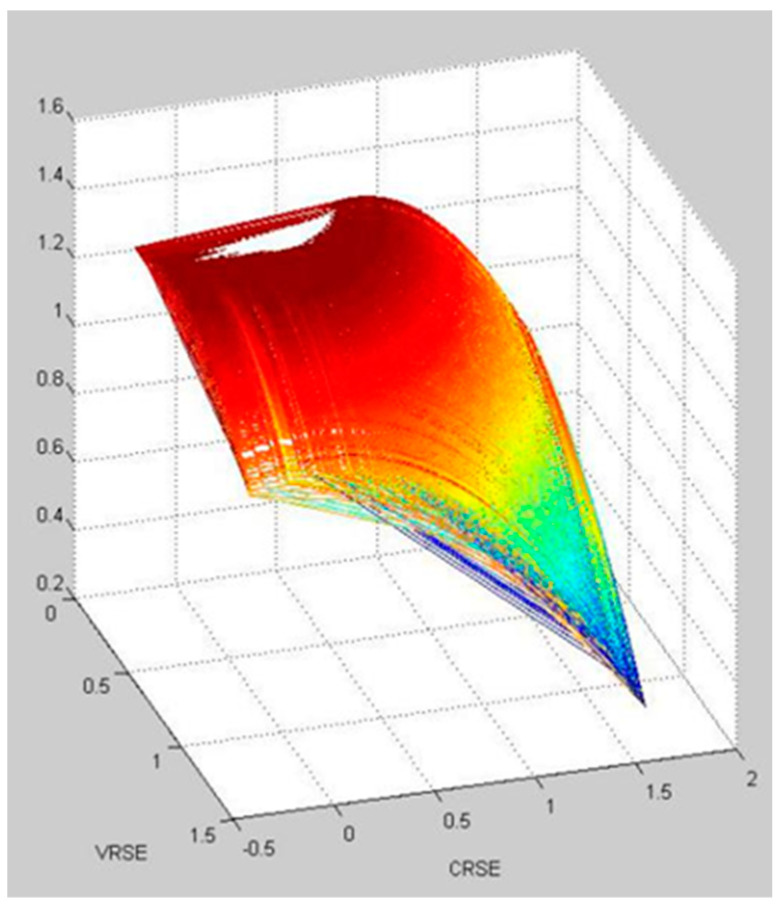
The visualization of simulation of the effect of Chinese outward FDI on wellbeing in Group 2.

**Figure 9 ijerph-20-05164-f009:**
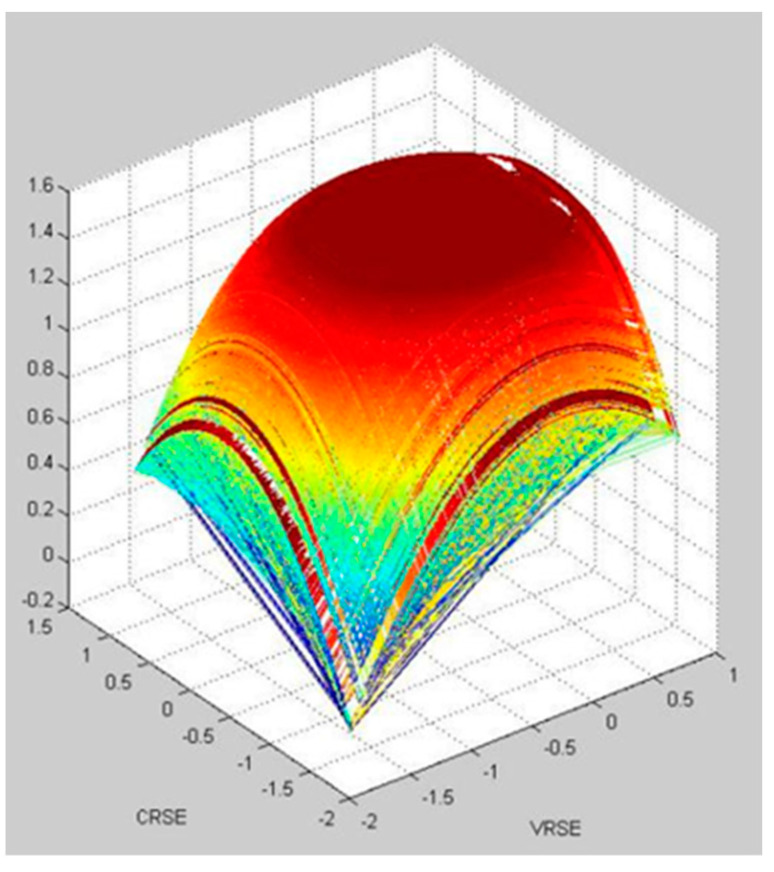
The visualization of simulation of the effect of Chinese outward FDI on wellbeing in Group 3.

**Figure 10 ijerph-20-05164-f010:**
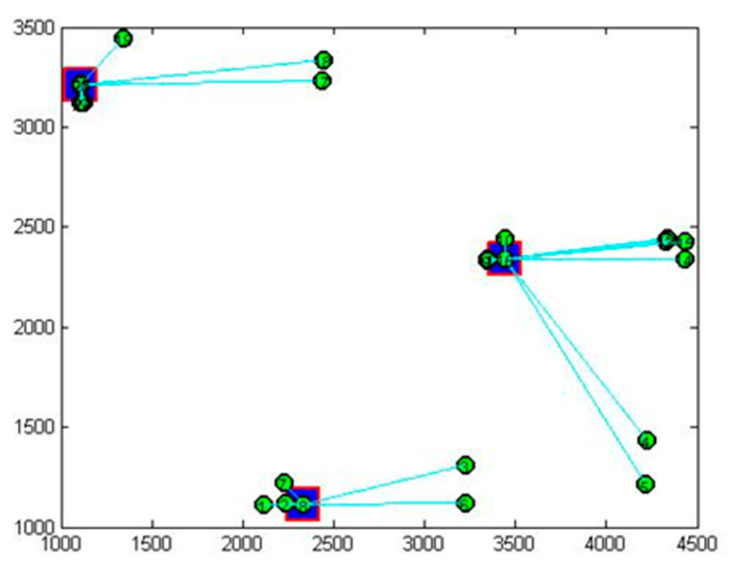
Key nodes in effect of Chinese outward FDI on wellbeing of OECD countries.

**Table 1 ijerph-20-05164-t001:** Theoretical framework in cross-case study about the effect of FDI on wellbeing.

Level	Theory	Effect	Code	Case
Social Level	Social Influence Theory [12]	Social Responsibility	Education, Libraries, and Hospitals,	“One Thousand Dream” program by Huawei in Central and Eastern Europe (CEE)
			Personnel	Midea provides consistent support of personnel functions of Toshiba
		Localization	Economics and Employment	Geely offers contributions to economics and employment in investment in Sweden
			Research	Geely innovative center in Europe serves the research of auto power
Organizational Level	Incentive Compatibility Theory [22]	Sustainability	Partnership and Cooperation	Lenovo developed partnerships in cooperation with 1500 companies in Italy, and cooperated with Italian NGO CINECA in High-Performance Computing (HPC)
		Survival	Continuous Development	Midea offers support to the continuous development of the Toshiba company from Japan, and the Shuanghui Group company maintains the operation and management group of Smithfield
Individual Level	Theory X vs. Theory Y [30]	Efficiency (Theory X)	Technological Support	Huawei offers technological support to digital management work in Latin America
			Efficiency	Lenovo enhanced efficiency in engineering design work for the Italian automobile manufacturer, Dallara
		Potential (Theory Y)	Training	The local government offers free training to employees in companies with Chinese investment in Germany
			Working Together	Changhong company works together with engineers from the Czech Republic to find solutions in assembly lines

**Table 2 ijerph-20-05164-t002:** Descriptive analysis of wellbeing in OECD countries for Chinese outward FDI.

	General Life Satisfaction	Housing	Income	Jobs	Community	Education
Mean	6.26	6.20	4.23	6.71	5.98	5.78
Std	1.56	1.71	2.82	2.37	2.38	2.33
Max	7.50	8.70	10.00	9.20	8.90	8.70
Min	0.67	1.09	0.30	1.51	1.87	0.60
Median	6.74	6.40	4.10	7.70	6.49	6.60
	Environment	Civic Engagement	Health	Happiness	Safety	Work–Life Balance
Mean	6.13	6.08	4.78	6.77	5.97	5.59
Std	2.06	2.22	3.22	2.56	2.55	2.66
Max	7.50	8.70	10.00	9.20	8.90	8.70
Min	0.67	1.09	0.30	1.51	1.87	0.60
Median	6.82	6.43	4.49	7.75	6.39	6.55
	FDI Flow	FDI Stock	Pop	GDP	Cultural Distance	
Mean	5.27	5.58	5.06	6.02	5.54	
Std	2.60	2.80	3.40	3.05	2.70	
Max	7.50	8.70	10.00	9.20	8.90	
Min	0.67	1.09	0.30	1.51	1.87	
Median	6.70	6.40	4.23	7.20	6.30	

**Table 3 ijerph-20-05164-t003:** Analysis of characteristics of country units by Tabu search.

Country 1	Country 2	Country 3	…	Country 24
Country 2	Country 1	Country 3	…	Country 24
Country 3	Country 2	Country 1	…	Country 24
…				
Country 24	Country 2	Country 3	…	Country 1

**Table 4 ijerph-20-05164-t004:** Coding system of immune algorithm in key node analysis.

VRS	CRS	FDIF	FDIS	Group	POP	GDP	Culture Dist
3	4	4	4	2	3	4	3
1	3	4	4	3	4	4	4

**Table 5 ijerph-20-05164-t005:** Interconnections of OECD destination countries for Chinese outward FDI using Tabu search.

Country	Group 1	VRS	CRS	Country	Group 2	VRS	CRS	Country	Group 3	VRS	CRS
Finland	1	0.82	0.83	Canada	9	0.95	1.45	The Netherlands	17	0.94	1.58
Ireland	2	0.92	1.03	Germany	10	1.02	1.50	Australia	18	0.82	1.49
Chile	3	1.04	1.06	France	11	1.03	1.48	USA	19	0.69	1.20
Mexico	4	1.08	1.18	Japan	12	1.06	1.44	Greece	20	0.41	0.70
Hungary	5	1.11	1.13	Brazil	13	1.12	1.48	Czech Rep	21	0.15	0.19
Austria	6	1.03	1.16	Italy	14	1.13	1.51	Switzerland	22	0.78	0.78
Sweden	7	0.92	1.20	Russia	15	1.14	1.90	Belgium	23	0.77	0.82
New Zealand	8	0.93	1.26	UK	16	1.04	1.75	Denmark	24	0.81	0.83

## Data Availability

The Chinese outward FDI data are obtained from the 2016 Statistical Bulletin of China’s Outward Foreign Direct Investment. The GDP and population of OECD countries in 2016 are obtained from the World Bank database. Wellbeing data and life relevant indexes are from the OECD better life index database. The cultural indexes were from the Geert–Hofstede database.

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
