# Peer review of "FDI and Wellbeing: A Key Node Analysis for Psychological Health in Response to COVID-19 Using Artificial Intelligence"

_ijerph, 2023, doi:10.3390/ijerph20065164_

Round 1

Reviewer 1 Report

This is an advanced research for public health administration in recovery from COVID-19 by using artificial intelligence. This research discussed the effective channels of FDI on well-being based on Theory X and Theory Y at individual level, incentive compatibility at organizational level, and social influence theory at social level. This research also studies the effect of FDI on well-being based on DEA model, and identifies the country groups based on Tab search. This research further develops the key node analysis based on immune algorithm, and discusses the practical solution to public administration for psychological health in recovery from COVID-19.

Limitations and Strength

Limitation

In literature study to effective channel of FDI on welling, it can be further revealed that the integrative function of FDI on well-being generate an organic system in public administration of psychological health in response to COVID-19 at individual level, organizational level, and social level, and this is an important contribution in further understanding the inter-disciplinary system of public health administration acquiring economic and psychological vision.

In further discussion about application of artificial intelligence to public administration for psychological health in response to COVID-19, it can be further revealed that this is an important study of public health administration in response to COVID-19 at global network. The super-efficiency DEA model offers support to study the effect of FDI from emerging market on well-being at destination countries, and Tabu search offers efficient solution in identifying the country groups in psychological health administration based on global heuristic solution. The key node analysis based on immune algorithm can be helpful to reveal the group characteristics based on study to key node countries in public health administration.

In development of practical implication, it can be further revealed that the practical solutions following the key node analysis based on artificial intelligence can be helpful to develop positive network externality in public health administration for country groups in response to COVID-19.

Strength

This research develops literature study to the effect of FDI on well-being, and offers a multi-level explanation to effect of FDI on well-being in supporting public psychological health in response to COVID-19 based on inter-disciplinary vision of economics and public health.

This research leads an inspiring analytic work in key node analysis for psychological health in recovery from COVID-19 by using artificial intelligence, and initiates a creative analytical framework to public psychological health in response to COVID-19 by AI technology based on involvement of FDI from emerging market.

This research also develops practical discussion to further administration of psychological health based on function of FDI on well-being. The advancement of practical solutions by support of AI technology can be helpful to further development of public administration for psychological health in response to COVID-19.

Comment on the methods, results and data interpretation.

This research has well explained the research methods and data sources. The effect of FDI on well-being is analyzed based on super-efficiency DEA model with development of efficiency measurement in constructing the frontiers. The country groups are identified by Tabu search, and the outperformance of this artificial intelligence technology in heuristic solution is explained. The key node analysis is further developed by immune algorithm, and the practical solution in public administration for psychological health in response to COVID-19 following the research findings is further discussed.

Detailed review report to the editor and authors

This is an initiative study that leads a new vision to public administration for psychological health in response to COVID-19 based on inter-disciplinary study of economics and public health by support of artificial intelligence technology. The discussion about multi-level effect of FDI on well-being reveals public administration for psychological health in recovery from COVID-19 at the individual, organizational and social level by involvement of FDI from emerging market. The advancement of super-efficiency DEA analysis enhanced the study to effect of FDI on well-being, and the outperformance of Tabu search aiming at global heuristic solution offers further support to identify the country groups in global network externality. The key node analysis based on immune algorithm offers a creative analytical framework to practical solution in public health of psychological health in response to COVID-19. This research contributes to further understanding to development of an organic public health system for psychological perspective in recovery from COVID-19 by support of artificial intelligence.

Reviewer 2 Report

Summarize the main findings of the study

This is an interesting research that reveals the effect of FDI on well-being by using artificial intelligence. The effect of FDI on well-being is discussed following literature study of channels at individual level, organizational level, and social level. The artificial intelligence is applied in analysis to effect of FDI on well-being in response to COVID-19 for psychological health perspective. The solution to further development of public administration system for psychological health in recovery of manufacturing and service work from COVID-19 is further discussed.

Limitations and Strength

Limitation

The inter-disciplinary contribution of this research based on economics and public health can be further discussed. It can be further revealed that this research is an important contribution to inter-disciplinary study of public health research, and this research offers support to psychological health in recovery of manufacturing and service network from COVID-19 based on artificial intelligence in computation science.

The analytical framework to public psychological health in response to COVID-19 by involvement of FDI with support of artificial intelligence can be further discussed. It can be further revealed that this research develops a leading study to effect of FDI on well-being by super-efficiency DEA model, and identifies the country groups by Tabu search. The analytical result is further involved in key node analysis for psychological health in response to COVID-19 by immune algorithm. The systematic analysis to well-being by vision of FDI from emerging market offers a new hint to further public administration of psychological health in recovery from COVID-19 based on artificial intelligence.

Strength

This research leads a creative study to effect of FDI on well-being by acquiring economic and health vision based on artificial intelligence technology. This research contributes to the inter-disciplinary study of public health in acquiring social science and computation science. This research further enhanced the literature in public health administration by involvement of economic issues, and initiated a new framework in public health study to effect of economic perspective based on artificial intelligence technology.

This research also develops an advanced analysis to public health administration in response to COVID-19 by support of artificial intelligence. The application of AI technology offers a new hint to explore practical solutions in public administration for psychological health in response to COVID-19 by acquiring economic perspective based on social factors. This is an important contribution that initiated a cutting-edge study of public health administration in response to COVID-19 by using artificial intelligence technology, and the artificial intelligence offers important support to advancement of the analytical framework in exploration of practical solutions.

Comment on the methods, results and data interpretation.

This research has rationale the research methodology and data sources. The advancement of super-efficiency DEA in measurement of effect on FDI on well-being is explained, and the advantage of Tabu search applied to identify the country groups is discussed. The key node analysis based on immune algorithm enhanced the study to practical solutions of public health administration in acquiring economics vision. The analytical framework of this research inspires a hint to development of inter-disciplinary public administration system for psychology health in response to COVID-19 by support of artificial intelligence technology.

Detailed review report to the editor and authors

This is an inspiring research work that initiates the study of public health administration in response to COVID-19 from psychological perspective by acquiring social science and computation science area. The development of organic system for public health administration in acquiring economics and psychological study contributes to inter-disciplinary literature of multi-level global governance at individual level, organizational level, and social level. The application of artificial intelligence in practical solution of public health administration from psychological perspective in response to COVID-19 leads to an advancement of analytical framework in global governance based on social science and computation science studies. The development of efficiency measurement by super-efficiency DEA and advancement of Tabu search in exploring the global heuristic solution to identification of country groups can be helpful to enhance the analytical framework in revealing the network externality of FDI on welling. The application of immune algorithm in key node analysis for psychological health in response to COVID-19 offers important support in development of practical solution for public health. This research can be helpful to further understand the development of social health ecosystem based on psychological behavior in response to COVID-19 by support of artificial intelligence.

Reviewer 3 Report

The article examines the impact of FDI from China on well-being in OECD countries.

At the macro level, investment is the basis for implementing policies for expanded reproduction, accelerated scientific and technological progress, and improved competitiveness and quality of life for citizens. Such a research topic is worthy of attention.

However, the article requires further elaboration for a number of reasons.

1. The article contains some unsubstantiated statements:

1.1 In lines 22-24, the authors argue that "Chinese outward FDI has an impact on well-being in destination countries, and this is an important issue for psychological health in response to COVID-19". However, the research section does not focus on the impact of psychological health and COVID-19 on the stated aim of the study.

1.2 The abstract also states (lines 27-28) that "this research has implications for global managers and business policymakers in shaping FDI policies to improve the psychological health in response to COVID-19 of destination countries". However, the text does not answer the question of what effect this research has had on global managers and business policymakers.

2. Lines 207-208; 238-240; state a well-known fact. Investment, by its very nature, is intended to create favourable conditions for the performance of expanded reproduction

3. The article is poorly structured. The research methods are not clearly distinguished in the text; the Authors refer to them both in the introduction (lines 72-80) and partially in the Data and Methods section.

4. If the subject of the study is OECD countries, why is not all OECD countries considered in this study?

5. The dataset presented in Figure 1 dates back to 2016, which in 2023 is a rather controversial argument in terms of the rate of economic change.

6. Table 1 is difficult to understand. The criteria for allocating each investment to the wealth levels proposed by the Authors are not clear. Moreover, the purpose of the 3 unnamed groups (Group 1, Group 2, Group 3), which are not described either in the table or in the text, is not at all clear. This table is also poorly suited to the purpose of the study, as it shows partial cases that have limited relevance to further research.

7. There is insufficient theoretical justification for the construction of Table 2. Before summarising the regularities, it would have been good to have initially researched the literature in more depth. The 3 sources cannot provide sufficient theoretical justification. The logical link between Tables 1 and 2 is broken; further explanation of the relationships is required. 

8. It is not possible to trace the links to references due to non-compliance with the rules of the list of references in the journal

9. The list of references should be improved, as in the provided list only 3 sources are dated after 2020. This calls into question the relevance of the trends discussed in the article.

Reviewer 4 Report

Dear authors,

I have been invited to review your paper. Attached you will find detailed comments and questions. In my view substantial work is required, before the editor can consider your paper for publication.

My comments cover several dimensions including the relevance and interest of the paper, structure of the paper, the completeness of the literature review, the justification of the methodological approach used, the choice of variables, the presentation of conclusions and their relevance. Moreover, I would recommend that a professional proofreader goes through the paper.

Thank you and regards,

Anonymous reviewer

Round 2

Reviewer 3 Report

The article is much better structured. Data that were of secondary relevance to the aim of the study have been removed. The Authors paid much more attention to the impact of psychological health on the stated aim of the study, which was required by the purpose of the article. Literature sources have been revised.

Reviewer 4 Report

Dear authors,

Thank you for the revised version of your paper.

Best regards,

Anonymous reviewer
